# Corticosteroid Treatment in Sydenham Chorea: A 27-Year Tertiary Referral Center Experience

**DOI:** 10.3390/children10020262

**Published:** 2023-01-31

**Authors:** Alberto Maria Cappellari, Greta Rogani, Giovanni Filocamo, Antonella Petaccia

**Affiliations:** 1Department of Neuroscience, Fondazione IRCCS Ca’ Granda Ospedale Maggiore Policlinico, 20122 Milan, Italy; 2Department of Pediatrics, Università degli Studi di Milano, 20122 Milan, Italy; 3Department of Pediatric Rheumatology, Fondazione IRCCS Ca’ Granda Ospedale Maggiore Policlinico, 20122 Milan, Italy

**Keywords:** Sydenham chorea, movement disorders, corticosteroids, treatment response

## Abstract

Objective: The purpose of this study was to investigate the effectiveness of corticosteroid therapy for children suffering from Sydenham chorea (SC). Methods: The design of the study was observational, retrospective and conducted at the single center of the Rheumatology Unit of Policlinic Hospital of Milan, Italy, from May 1995 to May 2022. All data about the patients were collected from medical records. Results: From a total of 59 patients enrolled in the study (44 females and 15 males; median age 9.3 years, range 7.4–10.6 years), 49 were eligible for primary outcome analysis (10 patients were excluded due to incomplete data). Overall, 75% of patients received steroid therapy, while the remaining cases were treated with symptomatic drugs, including neuroleptics and antiseizure drugs. We found that the duration of chorea was significantly shorter in patients treated with corticosteroids in comparison to those receiving symptomatic treatment (median time: 31 vs. 41 days, *p* = 0.023). Additionally, patients with arthritis at the onset of the disease had a longer duration of chorea than those without arthritis (median time 90.5 vs. 39 days, *p* = 0.02). We also found that chorea recurred in 12% of the patients and seemed to be linked to a younger age at onset (*p* = 0.01). Conclusions: The study suggests that corticosteroid therapy can lead to a faster resolution of SC when compared to neuroleptics and antiseizure drugs treatment.

## 1. Introduction

Chorea is a movement disorder characterized by a continuous flow of unpredictable involuntary movements [1,2]. Sydenham chorea (SC), a major manifestation of rheumatic fever (RF), is the most common form of acute chorea in children [3], usually developing 4 to 8 weeks after an episode of group A Beta-Hemolytic Streptococcal (GABHS) pharyngitis [4]. Treatment options for SC are still largely empirical [5]. The presence of antibodies reactive with neuronal tissue in the serum of patients with SC indicates a humoral mediated autoimmune condition [6,7]. Immunomodulatory therapies to shorten the duration of the illness and to prevent aggravations include corticosteroids, intravenous immunoglobulins (IVIG) and plasma exchange [8], with steroids having the strongest literature data [9].

Here we report the clinical experience with corticosteroid treatment in pediatric patients with SC referred to our tertiary referral center for RF over nearly three decades.

## 2. Materials and Methods

The design of the study was observational, retrospective and conducted at the single center of the Rheumatology Unit of Policlinic Hospital of Milan, Italy, from May 1995 to May 2022.

Clinical data of patients suffering from SC at disease onset and during a follow-up of at least 12 months were collected from medical records and registered in a case report form. All patients’ names and personal data were anonymized. 

The diagnosis of SC was established according to the revised Jones criteria [10]. For all patients, other causes of chorea were ruled out by appropriate investigations, including thyroid disorders, systemic lupus erythematosus, vasculitis and Wilson’s disease.

We required a positive throat culture or increased anti-streptolysin-O (ASO) titer to confirm a previous streptococcal infection, although their absence did not exclude the clinical diagnosis of SC. Indeed, 20–25% of acute chorea cases show no clinical or laboratory evidence of previous rheumatic disease [11].

The following patient data were collected: demographic data (age, sex, ethnic group), type of chorea (generalized or hemichorea), family history of RF, past medical history (including previous pharyngotonsillitis), presence of neuropsychiatric manifestations(emotional lability, irritability, difficulty in concentrating, mood disorders), presence of other major criteria of RF (carditis, arthritis, erythema marginatum, subcutaneous nodules), laboratory exams (C reactive protein, erythrocyte sedimentation rate, ASO titer, throat culture for group A streptococcus), echocardiography and brain MRI. Regarding the treatment, patients with SC were divided into two groups, the first one receiving steroid therapy and the second one receiving symptomatic treatment with neuroleptics or antiseizure drugs. The latency from onset of symptoms to diagnosis and start of treatment and the time needed for improvement and clinical remission were recorded, as well as the rate of recurrence.

Finally, we evaluated the compliance of secondary prophylaxis for GABHS, consisting of intramuscular penicillin G benzathine every 3 weeks for at least 5 years in patients without carditis and 10 years in those with carditis.

### Statistical Analysis

Categorical variables are described as number and percentage, while continuous variables as median and interquartile range (IQR). 

To assess the correlation between treatment and outcome parameters (time of improvement, time of remission and recurrence), Pearson chi-square test or Pearson correlation coefficient or the non-parametric Mann–Whitney test were used, when appropriate. *p*-values lower than 0.05 were considered significant.

The analyses were carried out leveraging SciPy, the open source scientific computing library for the Python programming language [12].

## 3. Results

A total of fifty-nine SC patients (44 females, 15 males), aged between 2 and 18 years (mean age 9.3 years) were enrolled for the study. Demographic characteristics and clinical features of the 59 patients with SC enrolled are illustrated below in Table 1. 

The median time from onset of symptoms to the diagnosis and start of treatment was 9 days (IQR = 3–15 days). Chorea’s improvement began after a median of 1 day after starting treatment (IQR 1–3,75 days), and clinical remission was achieved in 40 days (IQR 29–61 days). 

The relationship between type of therapy (steroid or symptomatic drugs) and symptoms’ duration was established in only 49 of the 59 patients enrolled, since the date of remission was not available in six patients and SC was diagnosed only after spontaneous remission in four patients. Among the 49 patients considered, 39 patients (79.5%) received steroid therapy as first line therapy or second choice when symptomatic therapy failed (neuroleptics, diazepam, valproate, respectively in nine, two, and one patients), while the remaining ten patients were treated only with symptomatic drugs (neuroleptics, valproate, diazepam respectively in eight, one, and one patients). Steroid therapy consisted of prednisone at the initial dose of 2 mg/Kg/day for 1–2 weeks, followed by tapering and discontinuation within 1–2 months. The duration of treatment with symptomatic drugs was variable, depending on the severity of chorea. Corticosteroids treatment was associated with transitory weight gain, while therapy with neuroleptics was complicated by unilateral foot dystonia in one patient, that disappeared after 6 months. 

Demographic characteristics and clinical features of the 39 patients treated with prednisone and of the ten patients treated with symptomatic drugs are showed below in Table 1.

The relationship between type of treatment and primary outcomes are detailed in Table 2 and represented in Figure 1. 

Although the interval for first improvement was not statistically different in the two groups, the time of remission was significantly shorter in patients treated with prednisone compared with those treated with symptomatic drugs (median time 31 vs. 41 days, *p* = 0.023). Furthermore, the group that received steroid presented an IQR of time for remission remarkably shorter (28 vs. 102 days), as shown in Figure 1.

Finally, the presence of arthritis at SC onset was associated with a longer duration of symptoms (median time 90.5 versus 39 days, *p* = 0.02), as shown in Figure 2.

More details about the demographic characteristics and the clinical features of the patients with and without arthritis are shown in Table 3.

Finally, we looked for a possible correlation between any demographic and clinical features and the occurrence of one or more recurrences of the SC. We found that the incidence of a recurrence was higher in the patients with a younger age at onset (median age at onset of patients with recurrence is 6.06 years versus 9.37 years, *p*-value 0.011), as shown in the Figure 3.

## 4. Discussion

The main result of our study was that corticosteroids induce faster remission of chorea in comparison to neuroleptics and antiseizure drugs. This result seems to agree with other studies reporting that corticosteroid therapy induces a rapid response if compared with standard therapy for chorea [13], placebo [6] or natural history of SC [14]. Different treatments of chorea have been suggested in patients with SC, including symptomatic drugs (antipsychotic, antiseizure drugs) and immunomodulatory therapy (steroids, IVIG, and plasma exchange) [15]. Both neuroleptics and antiseizure drugs are “off-label” therapies for SC, with no demonstrated efficacy established by randomized control studies [5]. Furthermore, SC may represent a risk factor for drug induced parkinsonism, and there is a need for caution when treating patients with SC with neuroleptics [16]. The pathophysiology of SC raises the possibility of immunomodulatory therapies being effective [2]. According to Barash et al. [14], we found that a short course of corticosteroids is associated with marked improvement of the involuntary movements, without side effects besides transitory weight gain. 

We wondered if the presence of carditis could have influenced the decision of starting steroid treatment in SC, but, contrary to our expectations, we found that eight patients with SC and carditis had never received steroids. All these patients were admitted before the 2006, i.e., before the first randomized double-blind study demonstrating the superiority of steroids over placebo to shorten the symptoms of SC [6].

Our study also suggested that corticosteroid treatment was not the only factor influencing the time of remission of chorea. Indeed, patients with arthritis at disease onset had a longer duration of chorea compared to patients without arthritis. In our opinion, this finding could be explained by the higher phase activity of the inflammatory disease in SC patients with arthritis compared to those without arthritis. 

Finally, our study showed that recurrence of chorea seems to be related only to a younger age at onset. Although SC is usually a monophasic disease [17], 13% to 42% of the cases recur, usually in the first few years [4,17,18,19,20]. Suggested risk factors associated with SC recurrence include irregular usage of antibiotic prophylaxis, failure to achieve remission within 6 months and prolongation of symptoms for more than 1 year [21]. Although some authors reported that penicillin G prophylaxis reduces recurrence compared with no prophylaxis [21,22,23], suggesting that most recurrences are due to re-exposure to streptococcal infection [19], a few reports mentioned that prophylactic treatment does not have any effect on recurrences [24,25]. Several patients with delayed recurrences of chorea that were not accompanied by active RF have been reported, suggesting that other unidentified triggers have induced an increase in the immunological reaction within the central nervous system [17]. However, the efficacy of immunomodulatory therapies in preventing recurrent episodes of SC remains controversial. Although Barash et al. suggested that commencement of corticosteroid treatment at an early stage could explain the lack of recurrence of SC [14], Paz et al. reported that recurrence rates were the same in children treated with steroids compared with placebo controls in a randomized, double-blind, parallel study [6]. Favaretto et al. reported that steroid therapy does not increase the risk of relapses but rather, on the contrary, could be effective in their prevention [13]. Recognizing that some authors suggest that recurrent episodes of SC may not be immune mediated, future studies of immunomodulatory therapies should attempt to correlate outcomes with measurements of proposed biomarkers [15].

Although we did not use standardized neuropsychiatric interviews, we noticed that a faster resolution of chorea in our patients treated with prednisone seemed associated with a better social reintegration at home and school. This personal observation could be consistent with the interesting finding reported by Moreira et al. that social phobia was more frequent in SC patients compared to the general population [26].

Our study has several limitations, including the retrospective nature of the study, the imbalance between patients treated with prednisone and those treated with symptomatic drugs (39 versus 10 respectively), and the lack of use of standardized chorea rating scales.

In conclusion, our study suggests that early treatment with corticosteroids in SC is associated with rapid improvement of the involuntary movement and could be considered as first-line therapy in all patients with SC unless contraindicated. We currently try to avoid symptomatic treatment with neuroleptics owing to the risk of drug induced parkinsonism, dystonia or both [1,27]. Larger, possibly comparative studies, using standardized assessment scales, are necessary if therapeutic decisions for SC are to be based on meaningful information [14,15].

## Figures and Tables

**Figure 1 children-10-00262-f001:**
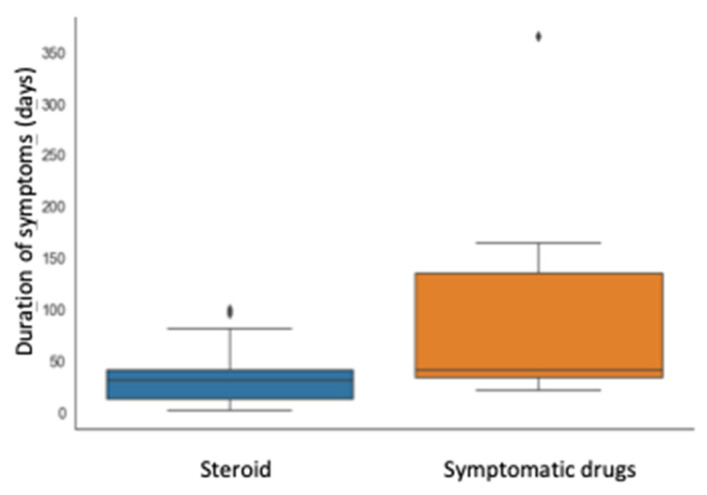
Correlation between therapy (prednisone or not) and symptom duration (days).

**Figure 2 children-10-00262-f002:**
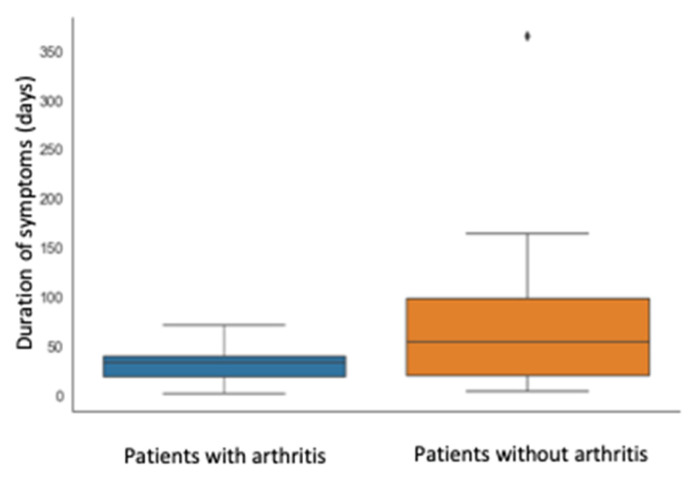
Correlation between arthritis (present or not) and symptom duration (days).

**Figure 3 children-10-00262-f003:**
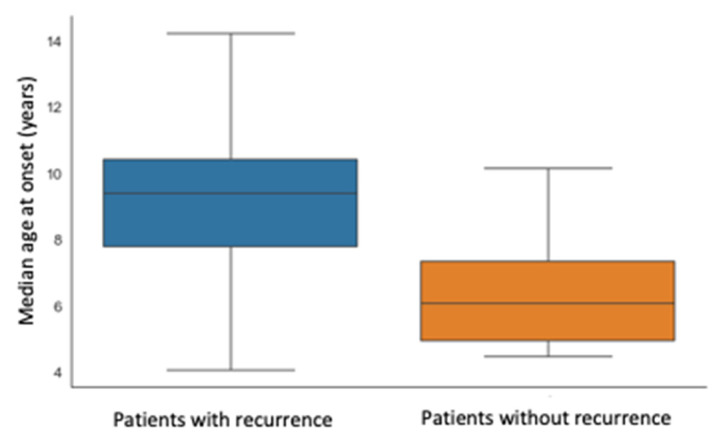
Correlation between the recurrence of SC and the median age of patients at the onset of the disease.

**Table 1 children-10-00262-t001:** Demographic characteristics and clinical features of the 59 patients with SC enrolled in the study, including 39 patients treated with prednisone and 10 patients treated with symptomatic drugs.

Demographic Characteristics	All Patients with SC (59)	Patients Treated with Prednisone (39)	Patients Treated with Symptomatic Drugs (10)
Median age (IQR)	9.3 years (3.2 years)	9.3 years (3.4 years)	8.2 years (1.8 years)
Sex	Female 44 (75%) Male 15 (25%)	Female 26 (66%) Male 13 (33%)	Female 10 (100%) Male 0 (0%)
Ethnic group	Caucasian 56 (95%) South American 3 (5%)	Caucasian 37 (95%) South American 2 (5%)	Caucasian 9 (90%) South American 1 (10%)
**Clinical features**			
Familiarity for RF	Positive 5 (8.4%)	Positive 3 (7.6%)	Positive 0 (0%)
Previous pharyngotonsillitis	Yes 38 (64.4%) No 19 (35.3%) Not available 2 (0.03%)	Yes 27 (69%) No 11 (28%) Not available 1 (3%)	Yes 7 (70%) No 3 (30%) Not available 2 (0.03%)
Throat culture at SC onset	Positive 15 (25.4%) Negative 31 (52.5%) Not done 13 (22%)	Positive 22 (56%) Negative 11 (28%) Not done 6 (16%)	Positive 6 (60%) Negative 3 (30%) Not done 1 (10%)
Hospitalization	No 8 (13.6%) Yes 51 (86.4%) Mean Time (days): 10.5	No 2 (5%) Yes 37 (95%) Mean Time (days): 19.7	No 7 (70%) Yes 30 (30%) Mean Time (days): 9.5
ASO titer (UI/mL)	957 (IQR: 1030)	966 (IQR: 1196)	1199 (IQR:268)
Arthritis	14 (24%)	11 (28%)	3 (30%)
Carditis	No 9 (15.3%) Yes 50 (84.7%), of which: One valve involved 37 (74%) Two valves involved 13 (26%)	No 3 (8%) Yes 36 (92%), of which: One valve involved 24 (66%) Two valves involved 12 (33%)	No 2 (20%) Yes 8 (80%), of which: One valve involved 7 (88%) Two valves involved 1 (12%)
Subcutaneous nodules	1 (0.02%)	1 (0.03%)	0 (0%)
Erythema marginatum	1 (0.02%)	0 (0%)	1 (10%)
Neuropsychiatric symptoms	14 (23%)	8 (20%)	2 (20%)
Hemichorea	14 (23%)	11 (28%)	1 (10%)
Irregular secondary prophylaxis for GABHS	4 (6%)	3 (77%)	1 (10%)

**Table 2 children-10-00262-t002:** The time for improvement, for remission (primary outcome) and the recurrence rate in the patients treated with steroid therapy among other drugs and only with symptomatic therapy.

	Patients Treated with Prednisone (39)	Patients Treated with Symptomatic Drugs (10)	*p*-Value
Time for improvement (IQR)	1 day	2 days	0.149
	(1–3)	(1–8)	
Time for remission (IQR)	31 days	41 days	**0.023**
	(13–41)	(33–135)	
Recurrence rate (%)	6 out of 39	0	0.433
	(15.4%)	(0%)	

**Table 3 children-10-00262-t003:** Demographic characteristics and clinical features of patients with and without arthritis.

Demographic Characteristics	Patients with Arthritis (14)	Patients without Arthritis (35)
Median age (IQR)	8.5 years (2.6 years)	9.3 years (3.2 years)
Sex	Female 10 (71%) Male 4 (29%)	Female 26 (74%) Male 9 (26%)
Ethnic group	Caucasian 12 (86%) South American 2 (14%)	Caucasian 33 (94%) South American 2 (6%)
**Clinical features**		
Familiarity for RF	Positive 0 (0%)	Positive 3 (8%)
Previous pharyngotonsillitis	Yes 12 (86%) No 2 (14%) Not available 0 (0%)	Yes 22 (63%) No 12 (34%) Not available 1 (3%)
Throat culture at SC onset	Positive 3 (21.5%) Negative 8 (57%) Not done 3 (21.5%)	Positive 9 (26%) Negative 20(57%) Not done 6 (17%)
Hospitalization	No 2 (14%) Yes 14 (86%) Mean Time (days): 12.2	No 3 (8%) Yes 32 (92%) Mean Time (days): 20.4
ASO titer (UI/mL)	1301.5 (IQR: 1165)	975 (IQR: 1128.5)
Carditis	No 1 (7%) Yes 13 (93%), of which: One valve involved: 10 (77%) Two valves involved: 3 (23%)	No 4 (11%) Yes 31 (89%), of which: One valve involved 7 (68%) Two valves involved 1 (32%)
Subcutaneous nodules	1 (21%)	0 (0%)
Erythema marginatum	1 (21%)	0 (0%)
Neuropsychiatric symptoms	3 (21.5%)	7 (20%)
Hemichorea	4 (26%)	8 (23%)
Irregular secondary prophylaxis for GABHS	12 (86%)	33 (94%)
Therapy	Prednisone 11 (79%) Symptomatic drugs 3 (21%)	Prednisone 28 (80%) Symptomatic drugs 7 (20%)

## Data Availability

The data presented in this study are available on request from the corresponding author.

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
