# Peer review of "Corticosteroid Treatment in Sydenham Chorea: A 27-Year Tertiary Referral Center Experience"

_children, 2023, doi:10.3390/children10020262_

Round 1
Reviewer 1 Report
The topic presented in the report can be interesting for clinicians and for future recommendations.
The title as well as the material and methods are presented adequately to the problem.
However, I have a few comments.
Some abbreviations remain unexplained.
Table 1. presented “Demographic characteristics and clinical features of the 49 patients with SC enrolled”. I recommend comparing the same data for the study groups: treated with steroids and symptomatic drugs. Likewise for patients with and without arthritis.
I suggest adding data on confirmation of previous streptococcal infection (RF diagnosis criterion).
"Carditis" (Table 1) does not show the number of patients.
Table 2. does not include the size of the study groups.
In my opinion, more data on the treatment should be provided, including the duration of steroid use and side effects. Likewise for symptomatic drugs. A description of the prophylaxis used in the study group would also be interesting.
“Furthermore, the group that received steroid presented an IQR of time for remission remarkably shorter (28 vs 102 days). “
I do not find these data in the table. I recommend explaining.
„Finally, our study showed that recurrence of chorea seems to be related only to a younger age at onset.”
I couldn't find this information in the results.
The study group not treated with corticosteroids was relatively small. In my opinion, the results do not justify such firm conclusions.
Author Response
Author's Reply to the Review Report (Reviewer 1)
- Some abbreviations remain unexplained.
The unexplained abbreviations in the text (SBEGA and TAS) have been corrected as group A Beta-hemolytic Streptococcal (GABHS) and anti-streptolysin-O (ASO).
- Table 1. presented “Demographic characteristics and clinical features of the 49 patients with SC enrolled”. I recommend comparing the same data for the study groups: treated with steroids and symptomatic drugs. Likewise for patients with and without arthritis.
Thank you for this suggestion. Table 1 was expanded to include the demographic characteristics and clinical features of the 39 patients treated with prednisone and 10 patients treated with symptomatic drugs. We added table 3 to summarize the data of the patients with and without arthritis.
- I suggest adding data on confirmation of previous streptococcal infection (RF diagnosis criterion).
We added the required data in the Materials and Methods as follows: “We required a positive throat culture or increased ASO titer to confirm a previous streptococcal infection, although their absence did not exclude the clinical diagnosis of SC. Indeed, 20-25% of acute chorea cases show no clinical or laboratory evidence of previous rheumatic disease.”
- "Carditis" (Table 1) does not show the number of patients.
The number of patients with and without carditis have been included in Table 1
- Table 2. does not include the size of the study groups.
The size of the study groups has been included in Table 2.
- In my opinion, more data on the treatment should be provided, including the duration of steroid use and side effects. Likewise for symptomatic drugs. A description of the prophylaxis used in the study group would also be interesting.
The treatment duration of both corticosteroids and symptomatic drugs has been added to the results as follows: “Steroid therapy consisted of prednisone at the initial dose of 2mg/Kg/day for 1-2 weeks, followed by tapering and discontinuation within 1-2 months. The duration of treatment with symptomatic drugs was variable, depending on the severity of chorea”.
The description of the prophylaxis used in our study group has been included in the materials and methods as follows: “Finally, we evaluated the compliance of secondary prophylaxis for GABHS, consisting of intramuscular penicillin G benzathine every 3 weeks for at least 5 years in patients without carditis and 10 years in those with carditis”.
- “Furthermore, the group that received steroid presented an IQR of time for remission remarkably shorter (28 vs 102 days). “I do not find these data in the table. I recommend explaining.
In the Table the IQRs (interquartile ranges) are expressed as 25th - 75th centile, while in the text they are expressed as an absolute difference between the 25th and the 75th centile, to make the comparison easier.
- Finally, our study showed that recurrence of chorea seems to be related only to a younger age at onset.” I couldn't find this information in the results.
The required information has been added to the results as follows: “Finally, we looked for a possible correlation between any demographic and clinical features and the occurrence of one or more recurrences of the SC. We found that the incidence of a recurrence was higher in the patients with a younger age at onset (median age at onset of patients with re-currence is 6.06 years versus 9.37 years, p-value 0.011), as shown in the figure 3.”
- The study group not treated with corticosteroids was relatively small. In my opinion, the results do not justify such firm conclusions.
You are right. We include your opinion in the final part of the discussion as follows: “Our study has several limitations, including the retrospective nature of the study, study, the imbalance between patients treated with prednisone and those treated with symptomatic drugs (39 versus 10 respectively), and the lack of use of standardized chorea rating scales”.
“In conclusion, our study confirms suggests……….. “
Reviewer 2 Report
This is a potentially interesting article, although it remains somewhat anecdotal. Several concerns have arisen before the reconsideration of its suitability for publication.
1. It is unclear what kind of novelty exists in this MS. This issue should be clarified.
2. The treatment duration of corticosteroids in the study cohort is unclear.
3. It is nice to reevaluate what kind of clinical factors are for the beneficial effects of corticosteroid treatment in patients. This issue should be clarified.
4. Some future perspectives in this field should be stated.
5. Also, adverse events of corticosteroid treatment should be represented.
Author Response
- It is unclear what kind of novelty exists in this MS. This issue should be clarified.
Although there is no novelty in this study, our work reinforces the idea of considering corticosteroids as first-line therapy in patients with Sydenham chorea. This concept is defined in the discussion as follows: “In conclusion, our study confirms that early treatment with corticosteroids in SC is associated with rapid improvement of the involuntary movement and could be considered as first-line therapy in all patients with SC unless contraindicated”.
- The treatment duration of corticosteroids in the study cohort is unclear.
The treatment duration of corticosteroids has been added in the results as follows: Steroid therapy consisted of prednisone at the initial dose of 2mg/Kg/day for 1-2 weeks, followed by tapering and discontinuation within 1-2 months”.
- It is nice to reevaluate what kind of clinical factors are for the beneficial effects of corticosteroid treatment in patients. This issue should be clarified.
A faster resolution of chorea in our patients treated with prednisone seemed associated with a better social reintegration at home and school. Although we did not use standardized neuropsychiatric examinations, this finding could be consistent with the data reported by Moreira et al. that “One interesting and new finding was the diagnosis of social
phobia in 24% of the patients, a frequency much higher than the expected for the general population”. This observation has been added in the conclusions as follows: “Although we did not use standardized neuropsychiatric interviews, we noticed that a faster resolution of chorea in our patients treated with prednisone seemed associated with a better social reintegration at home and school. This personal observation could be consistent with the interesting finding reported by Moreira et al. that social phobia was more frequent in SC patients compared to the general population (25).”
- Some future perspectives in this field should be stated.
This concept is defined in the discussion as follows: “Larger, possibly comparative studies, using standardized assessment scales, are necessary if therapeutic decisions for SC are to be based on meaningful information”.
- Also, adverse events of corticosteroid treatment should be represented.
Thanks for your important suggestion. The only side effect of corticosteroids in our population was represented by a transitory weight gain. This result is described both in the results as “Corticosteroids treatment was associated with transitory weight gain” and in the discussion as “According to Barash et al. (13), we found that a short course of corticosteroids is associated with marked improvement of the involuntary movements, without side effects besides transitory weight gain”.
Round 2
Reviewer 1 Report
I do not have more comments.
Reviewer 2 Report
The revised MS would be OK. The authors almost addressed my original concerns.